# Assembly Formation of P65 Protein, Featured by an Intrinsically Disordered Region Involved in Gliding Machinery of *Mycoplasma pneumoniae*

**DOI:** 10.3390/biom15030429

**Published:** 2025-03-17

**Authors:** Masaru Yabe, Takuma Toyonaga, Miki Kinoshita, Yukio Furukawa, Tasuku Hamaguchi, Yuhei O. Tahara, Munehito Arai, Katsumi Imada, Makoto Miyata

**Affiliations:** 1Department of Biology, Graduate School of Science, Osaka Metropolitan University, 3-3-138 Sugimoto, Sumiyoshi-ku, Osaka 558-8585, Japan; masayan0126@gmail.com (M.Y.); takuma.toyonaga.e3@tohoku.ac.jp (T.T.); kinoshita.miki.fbs@osaka-u.ac.jp (M.K.); tasuku.hamaguchi.c3@tohoku.ac.jp (T.H.); taharayuhei@omu.ac.jp (Y.O.T.); 2Institute of Multidisciplinary Research for Advanced Materials, Tohoku University, 2-1-1 Katahira, Aoba-ku, Sendai 980-8577, Japan; 3OMU Advanced Research Institute for Natural Science and Technology, Osaka Metropolitan University, 3-3-138 Sugimoto, Sumiyoshi-ku, Osaka 558-8585, Japan; 4Graduate School of Frontier Biosciences, Osaka University, 1-3 Yamadaoka, Suita 565-0871, Osaka, Japan; dezso2953@gmail.com; 5Department of Life Sciences, Graduate School of Arts and Sciences, The University of Tokyo, 3-8-1 Komaba, Meguro, Tokyo 153-8902, Japan; arai@bio.c.u-tokyo.ac.jp; 6Department of Macromolecular Science, Graduate School of Science, Osaka University, 1-1 Machikaneyama-cho, Toyonaka 560-0043, Osaka, Japan; kimada@chem.sci.osaka-u.ac.jp

**Keywords:** cell motility, protein structure, electron microscopy, small-angle X-ray scattering, adhesion, adhesin

## Abstract

*Mycoplasma pneumoniae* is a human pathogen that glides on host cell surfaces by a repeated catch and release mechanism using sialylated oligosaccharides. At a pole, this organism forms a protrusion called an attachment organelle composed of surface structures, including an adhesin complex and an internal core structure. To clarify the structure and function of the attachment organelle, we focused on a core component, P65, which is essential for stabilization of the adjacent surface and core proteins P30 and HMW2, respectively. Analysis of its amino acid sequence (405 residues) suggested that P65 contains an intrinsically disordered region (residues 1–217) and coiled-coil regions (residues 226–247, 255–283, and 286–320). Four protein fragments and the full-length P65 were analyzed by size exclusion chromatography, analytical centrifugation, circular dichroism spectroscopy, small-angle X-ray scattering, limited proteolysis, and negative staining electron microscopy. The results showed that P65 formed a multimer composed of a central globule with 30 and 23 nm axes and four to six projections 14 nm in length. Our data suggest that the C-terminal region of P65 is responsible for multimerization, while the intrinsically disordered N-terminal region forms a filament. These assignments and roles of P65 in the attachment organelle are discussed.

## 1. Introduction

*Mycoplasma pneumoniae* (*M. pneumoniae*), a species in the Mollicutes class, is a pathogen that induces human bronchitis and walking pneumonia, and it is characterized by its small cell and genome size and lack of a peptidoglycan cell wall [1,2,3,4]. This mycoplasma forms a membrane protrusion at cell poles, called an attachment organelle, that binds to solid surfaces such as human tissues. This protrusion causes gliding in the direction of the protrusion with a speed up to 1 μm (half of its cell length) per second [5,6,7,8,9,10]. This motility is critical for its infectivity, enabling the bacteria to translocate from the tips of bronchial cilia to the host cell surface [11,12]. Interestingly, this mechanism is unrelated to other known mechanisms of bacterial motility, including the gliding of *Mycoplasma mobile* (*M. mobile*) [5,13,14,15,16,17]. It is also unrelated to motor proteins that are widely distributed in eukaryotic motility [13]. The attachment organelle of *M. pneumoniae* is composed of an internal core and surface structures [5,18,19,20,21]. The surface structure is covered by an array of adhesin complexes that function as legs during gliding and another surface protein, P30, that localizes at the tip of the protrusion. The internal core structure can be divided into three parts: the terminal button, paired plates, and the bowl complex at the front of the cell. Although little is known about the gliding mechanism of *M. pneumoniae*, a possible model suggests that the force generated around the bowl complex is transferred through the paired plates to the adhesin complexes composed of P1, P40, and P90 proteins [22,23,24,25,26]. This results in the repeated catch and release of sialylated oligosaccharides fixed on the substrate surface [5,6,18,20,21,27,28,29,30].

P65 is composed of 405 amino acids and is essential for binding and gliding. It is also characterized by slow migration in a sodium dodecyl sulfate-polyacrylamide gel electrophoresis (SDS-PAGE) caused by the acidic and proline-rich regions (APR) including “DPN(Q)AY” sequences [31]. P65 localizes at the front end of the terminal button in the core complex. It localizes and stabilizes P30, which is a membrane protein at the tip of the bacteria that is essential for binding and gliding [32]. Possibly, P65 is necessary for connecting the internal core structure to the tip by interacting with HMW2, the major component of the core, and P30 [31,32,33,34]. Like other component proteins—HMW1, HMW3, P30 and P200—P65 is characterized by an APR and intrinsically disordered regions that are predicted to occupy more than half of the amino acid sequence [5,31,32,33,34].

In the present study, we isolated full-length P65 and four fragments of P65 using the *Escherichia coli* (*E. coli*) expression system and characterized them using fluid dynamics, optics, and electron microscopy (EM). Finally, we suggested the architecture of the P65 complex by describing the roles of the three regions of P65 in complex formation.

## 2. Materials and Methods

### 2.1. Sequence Analyses

Coiled-coil regions were predicted by COILS (version 2.2.1) [35]. Intrinsically disordered regions were predicted by the DISOPRED3 server (http://bioinf.cs.ucl.ac.uk/psipred/, accessed on 8 March 2025) [36]. α-helical regions were predicted by JPred4 (https://www.compbio.dundee.ac.uk/jpred/, accessed on 8 March 2025) [37]. Sequence similarity was determined using BLAST (https://blast.ncbi.nlm.nih.gov/Blast.cgi, accessed on 8 March 2025) against non-redundant protein sequences in the NCBI database (Appendix A) [38]. The sequences of homologs were aligned using T-COFFEE (https://tcoffee.crg.eu/apps/tcoffee/do:regular, accessed on 8 March 2025) [39].

### 2.2. P65 Protein Expression and Isolation

The codons of the *p65* gene (AGC04228) from the M129-B7 strain [40] were optimized for expression in *E. coli* BL21(DE3), synthesized (GenScript, Piscataway, NJ, USA), and inserted into pET-15b (Novagen, Madison, WI, USA) using NdeI and BamHI sites. The recombinant P65 preceded by a short peptide, MGSSHHHHHHSSGLVPRGSH, was expressed in *E. coli* with the addition of 1 mM IPTG at 37 °C for 3 h (Appendix A). The cells were harvested by centrifugation and suspended in 150 mM NaCl, 20 mM imidazole-HCl, 20 mM Tris-HCl (pH 8.0), and 0.1 mM phenylmethylsulfonyl fluoride (PMSF). Cells were disrupted using a US-600 sonicator (NIHONSEIKI, Tokyo, Japan) and centrifuged at 15,000× *g* for 30 min. The insoluble fraction was suspended in 150 mM NaCl, 20 mM imidazole-HCl, 20 mM Tris-HCl (pH 8.0), and 6 M urea. A soluble fraction obtained by the centrifugation was loaded onto a HisTrap HP 5 mL column, which was equipped in an AKTA pure 25 (Cytiva; Tokyo, Japan). The trapped protein was eluted with a linear gradient of imidazole ranging from 20 to 500 mM in 6 M urea, 150 mM NaCl and 20 mM Tris-HCl (pH 8.0).

The peak fractions that eluted in approximately 100 mM imidazole were collected, and the proteins were refolded by dialyzing three times against 150 mM NaCl, 1 mM EDTA, and 20 mM Tris-HCl (pH 8.0) at 4 °C. The collected fraction was loaded onto a HiLoad Superdex 200 26/600 size exclusion column (Cytiva) using an AKTA pure 25 (Cytiva) and collected fraction was for EM analyses and fragment B preparation. Thyroglobulin (669 kDa, Rs = 8.5 nm), ferritin (440 kDa, R_s_ = 6.1 nm), aldolase (158 kDa, R_s_ = 4.8 nm), conalbumin (75 kDa, R_s_ = 3.6 nm), and ovalbumin (44 kDa, R_s_ = 3.1 nm) were used as standards [41,42].

### 2.3. Preparation of P65 Protein Fragments

The P65 was digested with trypsin (Sigma-Aldrich, St. Louis, USA) (1/40 [wt/wt] to P65) in 150 mM NaCl and 20 mM Tris-HCl (pH 8.0) at 25 °C for 90 min (Appendix A). The reaction was quenched with the addition of 1 mM PMSF. The resultant fragments were analyzed by Peptide Mass Fingerprinting (PMF) (Appendix A). The digests were concentrated using an Amicon Ultra 30k spin filter (Millipore, Darmstadt, Germany) and loaded onto the size exclusion chromatography system. The mobile phase contained 150 mM NaCl and 20 mM Tris-HCl (pH 8.0) at a flow rate of 2.5 mL/min at 25 °C. The peak fractions of fragments A and C found at approximately 200 and 220 mL were collected, respectively. The fraction at 200 mL was used as fragment A (Appendix A). To purify fragment C, the peak fraction at 220 mL was loaded onto a HiTrap Q HP (Cytiva). The trapped protein was eluted with a linear gradient of NaCl ranging from 0 to 1 M in 20 mM Tris-HCl (pH 8.0). The peak fractions found around 80 mM NaCl were collected and used as fragment C (Appendix A).

To prepare fragment B, refolded P65 was digested with endopeptidase Asp N (Wako Pure Chemical Industries, Osaka, Japan) (1/40 [wt/wt] to P65) in a buffer consisting of 150 mM NaCl and 20 mM Tris-HCl (pH 8.0) at 25 °C for 16 h (Appendix A). The enzyme reaction was quenched with the addition of 1 mM EDTA. The digests containing fragment B were used for negative staining EM.

The recombinant fragment D preceded by the short peptide, MGSSHHHHHHSSGLVPRGSH, was expressed in *E. coli* as described for full-length P65 (Appendix A). Cells were harvested by centrifugation and resuspended in 150 mM NaCl, 20 mM imidazole-HCl, 20 mM Tris-HCl (pH 8.0), and 1 mM PMSF. Cells were disrupted using the sonicator and centrifuged at 15,000× *g* for 30 min. The soluble fraction was loaded onto a HisTrap HP column. The trapped protein was eluted with a gradient of imidazole ranging from 20 to 500 mM in 150 mM NaCl and 20 mM Tris-HCl (pH 8.0). Fragment D was recovered in fractions at 100 mM imidazole.

The MALTI-TOF mass spectrometry including PMF and other biochemical analyses were performed as previously described [28].

### 2.4. Size Exclusion Chromatography and Analytical Centrifugation

Isolated fragment D at 1 mg/mL was applied to a HiLoad Superdex 200 26/600 (Cytiva) equipped to an AKTA pure 25 (Cytiva). The mobile phase was 150 mM NaCl and 20 mM Tris-HCl (pH 8.0) at 2.5 mL/min at 25 °C. Ferritin (440 kDa, R_s_ = 6.1 nm), aldolase (158 kDa, R_s_ = 4.8 nm), conalbumin (75 kDa, R_s_ = 3.6 nm), and ovalbumin (44 kDa, R_s_ = 3.1 nm) were used as standards [41,42].

P65 and fragments A, C, and D were analyzed by analytical ultracentrifugation in 150 mM NaCl and 20 mM Tris-HCl (pH 8.0) as reported previously [43,44].

### 2.5. Far-Ultraviolet (UV) Circular Dichroism (CD) Spectroscopy

Far-UV CD spectra of the isolated P65 were measured as described previously [28] in 150 mM NaCl and 20 mM Tris-HCl (pH 8.0) with and without 6 M GdnHCl. The contents of protein secondary structures were estimated from the CD spectra by K2D3 (https://cbdm-01.zdv.uni-mainz.de/~andrade/k2d3/, accessed on 8 March 2025) [45].

### 2.6. Small-Angle X-Ray Scattering (SAXS) Measurements

SAXS measurements were performed at the beamline (BL)-10C at the Photon Factory of the High Energy Accelerator Research Organization (KEK), Tsukuba, Japan. The camera length was approximately 2 m and calibrated by powder diffraction using silver behenate with the FIT2D software [46]. Samples were loaded into a mica-windowed cuvette with a 1 mm path length and were irradiated with a monochromatic X-ray beam (1.488 Å). The concentrations of fragment A were 1 and 2 mg/mL. The temperature in the cuvette was maintained at 25 °C using a circulating water bath. Scattering images were acquired with a PILATUS3 300 KW detector (DECTRIS Ltd., Baden, Switzerland) with approximately 10 × 1 s exposures. Scattering data (*Q*) were collected from 0.015 to 0.25 Å^−1^ where *Q* = 4π sin (*θ*/*λ*) (*λ*, wavelength; 2*θ*, scattering angle). Scattering of the blank condition (buffer alone) was measured before and/or after measuring protein samples.

### 2.7. SAXS Data Analysis

Circular averaging of two-dimensional scattering data was performed with FIT2D [46]. The blank was subtracted from the scattering data of the protein solution to obtain the scattering profile of the protein molecule, *I*(*Q*). The corrected scattering data were binned per 4 data points to increase the signal-to-noise ratio of the data. The radius of gyration (*R_g_*) and the zero-angle scattering intensity (*I*(0)) were obtained based on the Guinier approximation within the Guinier region (*R_g_**Q* < 1.3) [47]:$$\ln I(Q)=\ln I(0)-\frac{R_{g}^{2}}{3}Q^{2}$$

The *R_g_* at zero protein concentration was obtained by extrapolation of the *R_g_* values at 1 and 2 mg/mL. The scattering curve at zero protein concentration was obtained by extrapolation of *I*(*Q*)/*c* at each *Q*, where *c* is the protein concentration.

### 2.8. Negative Staining EM

Proteins at 40 µg/mL were stained for 1 min with 2% uranyl acetate (w/v) on a carbon coated and glow discharged grid, air-dried, and observed by a JEOL, JEM-1010 transmission EM (Tokyo, Japan) at 80 kV equipped with a FastScan-F214 (T) charge-coupled-device (CCD) camera (TVIPS, Gauting, Germany), as previously described [48]. Images were analyzed with ImageJ v1.37 (https://imagej.net/ij/, accessed on 8 March 2025), as described previously [49]. Particle images were binarized in uniform manner, and their dimensions were measured as oval.

## 3. Results

### 3.1. P65 Can Be Divided into Three Regions Based on Amino Acid Sequence

To determine the sequence identity and similarity of P65 with other proteins, we performed a BLAST search against non-redundant protein sequences. The results showed that P65 has significant identity (41%) and similarity (50%) with MG_217 from *Mycoplasma genitalium* (*M*. *genitalium*) (Appendix A). The N-terminal region is composed of residues 1–177 and contained some gaps in the alignment. In the distantly related species, *Mycoplasma gallisepticum*, PlpA (*MGA1199*) is a suggested homolog of P65. PlpA has a longer amino acid sequence composed of 449 residues with several insertion sequences throughout the alignment with P65 [7]. The coiled-coil regions, intrinsically disordered regions, and secondary structures of P65 were predicted using COILS, DISOPRED3, and JPred4, respectively. COILS analysis indicated that residues 226–247, 255–283 and 286–320 were likely to form coiled-coil structures (Figure 1A upper panel) [50]. DISOPRED3 suggested that residues 1–217 are likely to be intrinsically disordered (Figure 1A lower panel) [36]. JPred4 suggested that P65 is composed of approximately 14% α-helical regions as indicated by red bars (Figure 1B) [37]. Based on these analyses, P65 can be divided into three regions: Ι, II and III. Region I is composed of the N-terminal 217 residues, including the APR, and is predicted to be intrinsically disordered (Figure 1A lower panel). Region II consists of residues 218–320 and is expected to contain three coiled-coil segments (residues 226–247, 255–283, and 286–320) (Figure 1A upper). Region III consists of residues 321–405 and may form a domain containing α-helices (Figure 1B).

### 3.2. P65 Forms a Large Complex

The full-length P65 protein fused with an N-terminal histidine tag was expressed in *E. coli*. As P65 was insoluble, it was solubilized and isolated with 6 M urea by Ni^2+^-affinity chromatography, refolded by dialysis, and then purified by size-exclusion chromatography (Appendix A). Refolded P65 eluted at a position corresponding to a weight much larger than the 49.2 kDa mass calculated from the amino acid sequence by size filtration, suggesting that P65 forms a multimer (Figure 2A). The refolding of P65 was confirmed by far-UV CD spectroscopy (Figure 2B). The analysis of the spectrum with the algorithm, K2D3, showed that the refolded P65 had 13.7% α-helical content and 29.4% β-strand content. These data were in agreement with the secondary structure prediction by JPred4 [45], suggesting that P65 is a complex of folded proteins rather than an aggregate of unfolded ones.

The multimerized P65 was further analyzed by determining its sedimentation velocity (Figure 2C, Table 1). The sedimentation coefficient distribution profile of P65 showed a single sharp peak at a sedimentation constant (S20.w) of 22.2. The derived molecular mass of the peak was approximately 3.10 × 10^6^ Da, suggesting that P65 forms a multimer with more than 60 subunits. Next, we observed the P65 assembly by negative staining EM. We found many star-like particles comprised a central spheroid core with axes of 30 ± 3 and 23 ± 3 nm and four to six projecting spines with lengths of 14 ± 2 nm (Figure 2D,E). The number of the observed particles positively correlated with P65 concentration. Thus, we concluded that P65 forms a star-like complex. Next, we prepared and analyzed four fragments of P65 and suggested the architecture of the complex based on data of the three regions.

### 3.3. Region I (Mostly Corresponding to Fragment A) Is Intrinsically Disordered

Fragment A comprising residues 1–209 was prepared by treating full-length P65 with trypsin (Figure 1B and Appendix A), because fragment A does not contain lysine or arginine residues sensitive to trypsin (Appendix A). The MALDI-TOF analysis showed 22,132 Da as the total mass of isolated protein, which was in agreement with the theoretical mass of Fragment A, 23,868 Da. We examined the oligomerization state of fragment A using size exclusion chromatography and analytical ultracentrifugation. Fragment A eluted as a single peak in the chromatography with an estimated Stokes radius of 3.4 nm (Figure 3A, Table 1). The sedimentation coefficient distribution profile showed a single symmetrical peak with a sedimentation constant of 0.97 (Figure 3B). The derived molecular mass of the peak was 25.7 kDa, which agreed with the theoretical mass of fragment A (23,868 Da), suggesting that fragment A is a monomer in solution. The sedimentation velocity analysis using SEDFIT [44] suggested that fragment A can adopt a thin cylindrical shape of 48.7 nm in length and 1.1 nm in diameter. The cylindrical shape and sizes were consistent with the faster elution speed observed by size exclusion chromatography compared to spherical protein markers.

To examine the secondary structure of region I, the far-UV CD spectra of fragment A were measured in the presence or absence of 6 M guanidine hydrochloride (GdnHCl). The CD spectrum of fragment A in the absence of GdnHCl showed a minimum at 198 nm, indicating that fragment A is disordered in solution (Figure 3C) [51]. However, a slight shoulder was observed between 210 and 230 nm. Furthermore, the intensity in this region became closer to zero when 6 M GdnHCl was added. These results suggest that fragment A has a residual secondary structure in solution, as has often been observed in intrinsically disordered regions [52]. This structure is unlikely to be derived from the short peptide “MGSSHHHHHHSSGLVPRGSH” fused at the N-terminus, because this sequence is predicted to be fully disordered by ColabFold [53] and has been reported to have disordered structures when fused to a globular protein (PDB ID: 1UPI) [54]. The peak at 230 nm that was observed in the presence of GdnHCl was likely due to the exciton coupling of tyrosine residues [55], since there are 35 tyrosine residues in fragment A, including nine YY sequences (Appendix A).

To estimate the size and shape of fragment A, we carried out SAXS measurements. The Kratky plot of fragment A shows a plateau (Figure 3D left). If a protein adopts a compact globular shape, the Kratky plot should show a clear peak. On the other hand, if a protein has an extended and unfolded structure, the plot has no peak and has a plateau [56,57]. Therefore, fragment A has an extended and unfolded structure in solution. The *R_g_* obtained from the Guinier plots [57] was 37.0 ± 1.0 Å and 32.1 ± 0.5 Å with 1 and 2 mg/mL solutions of fragment A, respectively (Figure 3D right). The decrease in the *R_g_* value at the higher protein concentration is probably due to an interparticle interference effect as indicated by a downward curvature toward smaller *Q* in the scattering curve at 2 mg/mL [57]. This effect was removed by extrapolating the scattering curves to zero protein concentration. The Guinier plots of the scattering curve of fragment A at zero concentration gave an *R_g_* of 41 ± 3 Å. According to the scaling relationship between the number of residues and the *R_g_* for chemically unfolded proteins, the *R_g_* of the fully unfolded state of a 209-residue protein is estimated to be 47.0 Å [58]. This value is slightly larger than the *R_g_* of fragment A, indicating that fragment A is slightly more compact than the fully unfolded state. In combination with the far-UV CD spectra of fragment A, we concluded that fragment A is intrinsically disordered but contains partially folded regions.

### 3.4. Region I (Fragment A) Forms the Flexible Spines and Regions II–III (Fragment B) Form Core of the P65 Star-like Particle

EM analysis demonstrated that P65 assembles into a star-like complex with four to six flexible spines. Sedimentation velocity analysis suggested that fragment A adopts a thin rod shape structure. Therefore, region I possibly forms the flexible spines. To examine this hypothesis, we focused on the fragment lacking region I. P65 contains 28 aspartic acid residues in region I and 10 in regions II and III (Appendix A). Thus, the endopeptidase, Asp N, was used for the limited digestion of P65. After 16 h incubation with AspN, an intense band of approximately 25 kDa was observed by SDS-PAGE (Appendix A). The molecular mass of the digestion product was determined to be 26.3 kDa by mass spectrometry. This value was in good agreement with the theoretical mass of residues 185–405 (fragment B, 26,279 Da; Figure 1B). The digestion product was further analyzed by PMF and was confirmed to cover at least residues 212–405, suggesting that the product was fragment B (Appendix A).

We then observed fragment B by negative staining EM and found spheroid particles with significantly smaller numbers of spines than the full-length protein (Figure 4). These observations suggest that the flexible spines are formed by region I and the spheroid core is composed of regions ΙΙ and ΙΙΙ.

### 3.5. Region II (Fragment C) Forms a Trimer

Next, we focused on fragment C. We treated P65 with trypsin, isolated fragment C and confirmed by PMF (Appendix A). We characterized the oligomerization state of fragment C using ultracentrifugation. The sedimentation coefficient distribution profile of fragment C showed a single peak with a sedimentation constant of 1.3 (Figure 5, Table 1). The derived molecular mass of the peak was 25.6 kDa, which agreed with the theoretical mass of a trimer of fragment C (25,893 Da).

The trimer formation of region II was supported by prediction using AlphaFold2 [53]. The predicted structure showed a parallel three-stranded coiled coil (Appendix A). The prediction accuracy of the trimer, as indicated by the predicted local distance difference test (pLDDT) scores, was very high for many residues of region II. In contrast, the dimer and tetramer predictions showed low pLDDT scores, supporting that region II forms a trimer.

### 3.6. Regions I–II (Fragment D) Forms a Trimer

We have shown that regions II and III form the spheroid core of the P65 star-like complex. To clarify the role of regions II and III in forming the spheroid core, we prepared a fragment composed of regions I and II (fragment D, residues 1–320; Figure 1B). Fragment D was expressed with an N-terminal histidine tag in *E. coli* and isolated by Ni^2+^-affinity chromatography followed by size exclusion chromatography (Appendix A). Fragment D eluted as a single peak with an estimated Stokes radius of 5.9 nm and a molecular mass of 419 kDa, suggesting it forms a spherical multimer (Figure 6A, Table 1). The oligomerization state of fragment D was further analyzed by ultracentrifugation (Figure 6B, Table 1). The sedimentation coefficient distribution profile of fragment D showed a sharp single peak with a sedimentation constant of 2.6. The derived molecular mass of the peak was 105 kDa, which agreed with the theoretical mass of the fragment D trimer (117,240 Da). This result indicates that region II is required for trimer formation and region III is needed to assemble the trimers into the star-like complex.

## 4. Discussion

### 4.1. Implication for the Star-like Complex Formation

Figure 7 shows the structure of the molecule and its complex based on the experimental results. The complex observed by EM was composed of a central globule with axes of 30 and 23 nm and four to six filaments approximately 14 nm in length (Figure 2D,E). Based on the structural prediction from the amino acid sequence, P65 can be divided into three regions: region Ι (the N-terminal intrinsically disordered region composed of residues 1–217), region ΙΙ (the coiled-coil region composed of residues 218–320), and region ΙΙΙ (the C-terminal region composed of residues 321–405) (Figure 1B). The EM results showed that the filamentous part of refolded P65 disappeared after digestion with the endopeptidase, Asp N (Figure 4). The digested region was assigned to residues 1–185 by PMF, suggesting that the filamentous and central parts are composed of residues 1–185 and 186–405, respectively. Analysis of the amino acid sequence suggested that region I is intrinsically disordered (Figure 1A), which was supported by the observation that 6 M GdnHCl did not significantly alter the CD spectrum of fragment A (Figure 3C). This conclusion was also supported by the analytical ultracentrifugation and the SAXS analyses (Figure 3B,D).

The central part of the structure is likely composed of regions II and III. Region II forms a trimer based on the analytical ultracentrifugation results that showed trimerization for both fragments C and D (Figure 5 and Figure 6B). Analytical ultracentrifugation indicated that full-length P65 formed a multimer weighing 3.10 × 10^3^ kDa or more, suggesting that region III is responsible for its multimerization.

### 4.2. Assignment of the Star-like Complex in the Gliding Machinery

Figure 8A shows the location and the role of P65 in the attachment organelle that is composed of the internal core and surface structures [5,18,19,20,21,59,60,61].

In a previous study, P65 localization was determined by fluorescent protein tagging and immunoelectron microscopy [21,62]. P65 was found near the front end of the terminal button in the core of the attachment organelle. The schematic diagram of P65 obtained in the present study might overlay well with the image of the terminal button visualized through triton extraction and centrifugation (Figure 8B) [21].

P30, a membrane protein essential for adhesion and gliding, localizes at the front end of the *M. pneumoniae* surface [33,63]. Region III of P65 likely interacts with P30 at the front end of the cell, because this region is indispensable for the stability of both proteins [32].

HMW2, the major component of paired plates, is also essential for the stability of P65 [64,65]. P65 likely interacts with other proteins through region I, since intrinsically disordered regions are commonly found in protein–protein interactions [52]. It is possible that P65 bundles P30 by forming multimers and connects the front end of the paired plates’ core structure to the front end of the attachment organelle. This model is consistent with previous observations and our suggested mechanism [32]. The force for gliding may be generated around the bowl complex, which is the back side of the internal core structure [5,20]. The force from the back side along the internal core could be transmitted to the adhesin complexes through the stable P65 complex and P30 cluster.

## 5. Conclusions

We clarified the structural outline of the P65 protein, a component of the internal structure for *M. pneumoniae* binding and gliding, by using a recombinant protein. In the future, this information would be useful to assign the P65 protein into the whole image of the attachment organelle, probably based on the cryo-EM of a cell and isolated organelle, and discuss the role of P65 protein in binding and gliding mechanisms.

## Figures and Tables

**Figure 1 biomolecules-15-00429-f001:**
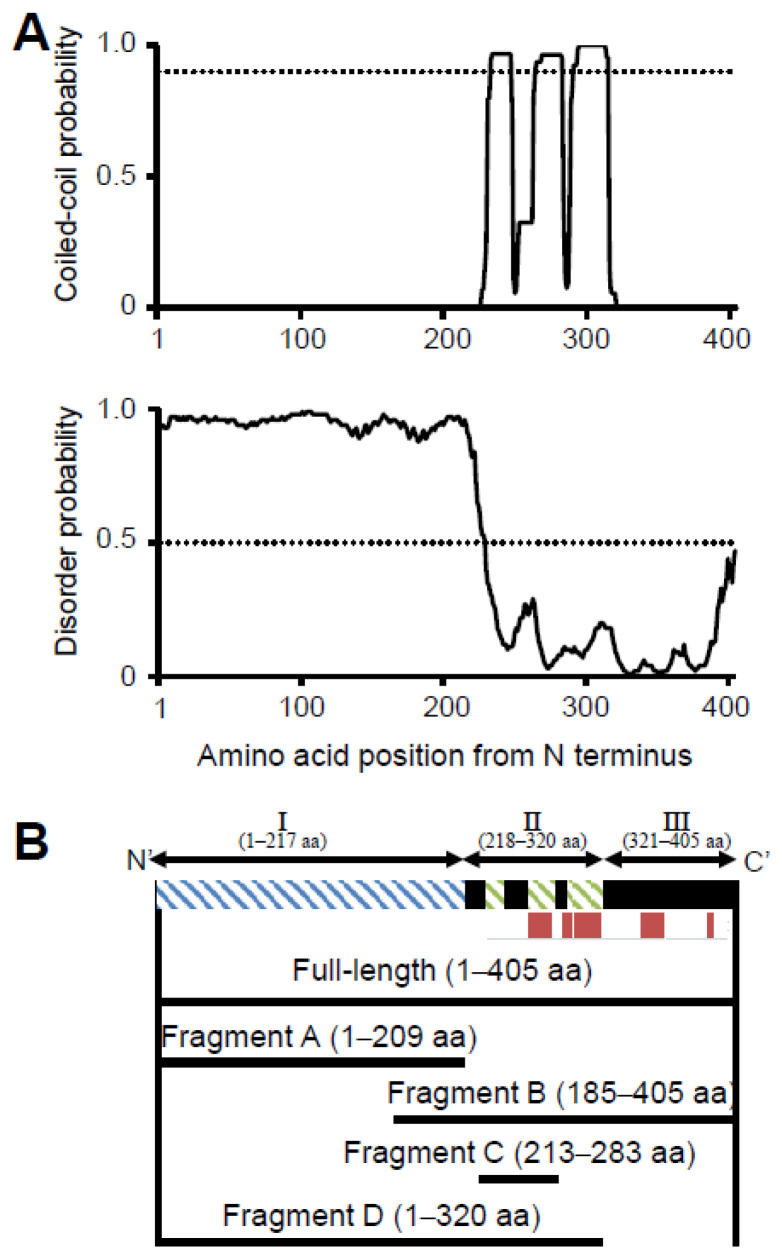
Construction of fragments for structural analyses of P65 based on its amino acid sequence. (**A**) Sequence analyses. Upper: Coiled-coil regions were predicted by COILS [25]. The probability was scanned against the amino acid sequence using 14 residue windows. The probability at 0.9 is marked by a dotted line. Lower: Intrinsically disordered regions were predicted by the DISOPRED3 server [36]. The disorder probability versus the amino acid sequence was plotted with a mark at 0.5 indicated with a dotted line. (**B**) Schematic representation of the P65 amino acid sequence (**upper**) and fragments analyzed in this study (**lower**). The green and blue hatch marks show the predicted coiled-coil and intrinsically disordered regions, respectively. The full-length sequence was divided into three regions: the intrinsically disordered region (region I), the coiled-coil region (region II), and the C-terminal region (region III). APR composed of the amino-acid sequence “DPN(Q)AY” and an α-helix predicted by the JPred4 server are denoted with blue and red lines, respectively. Four fragments and the full-length protein used in this study are shown by solid lines.

**Figure 2 biomolecules-15-00429-f002:**
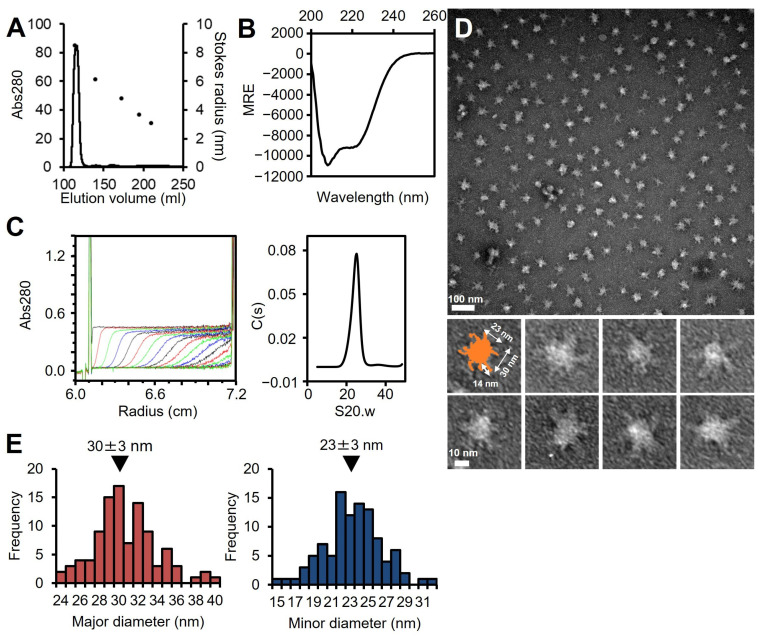
Analysis of full-length P65. (**A**) Size exclusion chromatography at a flow rate of 2.5 mL/min. Standard proteins with Stokes radii of 8.5, 6.1, 4.8, 3.6, and 3.1 nm were used (black dots). (**B**) Far-UV CD spectrum of isolated P65 at 0.2 mg/mL in 20 mM Tris-HCl pH 8.0. MRE denotes the mean residue ellipticity. (**C**) Analytical ultracentrifugation. Left: The P65 protein was analyzed by ultracentrifugation and scanned for absorbance at 10-minute intervals. Twenty representative traces with 50-minute intervals are shown. The horizontal axis shows the position from the rotational center. Right: The distribution of C(s) plotted against S20.w. The results are summarized in Table 1. (**D**) Negative staining electron microscopy. (**E**) Distributions of the major and minor axes for the central globular region. The average is denoted by a black triangle.

**Figure 3 biomolecules-15-00429-f003:**
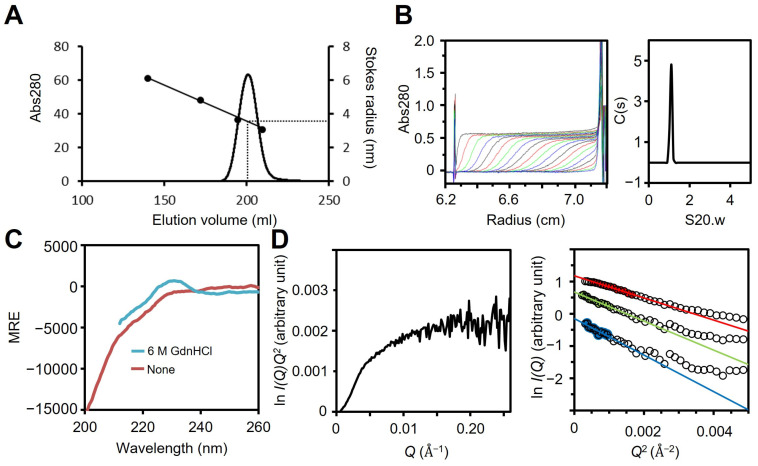
Analysis of fragment A. (**A**) Size exclusion chromatography at a flow rate of 2.5 mL/min. Standard proteins with Stokes radii of 6.1, 4.8, 3.6, and 3.1 nm were used (black dots). (**B**) Fragment A was analyzed as described for Figure 2C. The results are summarized in TABLE 1. (**C**) CD spectra with (cyan) and without (red) 6 M GdnHCl. In the presence of 6 M GdnHCl, measurement of the CD spectrum below 212 nm was prevented by the large absorption of GdnHCl. (**D**) SAXS analysis. Left: Kratky plot using P65 at 2 mg/mL. Right: Guinier plots using P65 at 1 (green) and 2 (red) mg/mL. The data obtained by the extrapolation of these curves to zero concentration are shown in blue. For clarity, this curve is shifted to a smaller value of ln *I*(*Q*) by −1.0. The continuous lines show the Guinier fitting, and the filled circles show the data used for the Guinier fitting.

**Figure 4 biomolecules-15-00429-f004:**
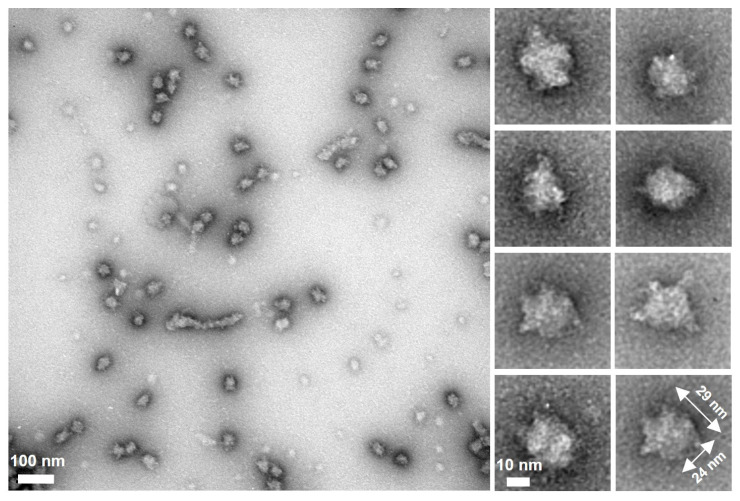
Analysis of fragment B. Isolated fragment B observed by negative staining EM. A field and selected regions are shown in the left and right panels, respectively.

**Figure 5 biomolecules-15-00429-f005:**
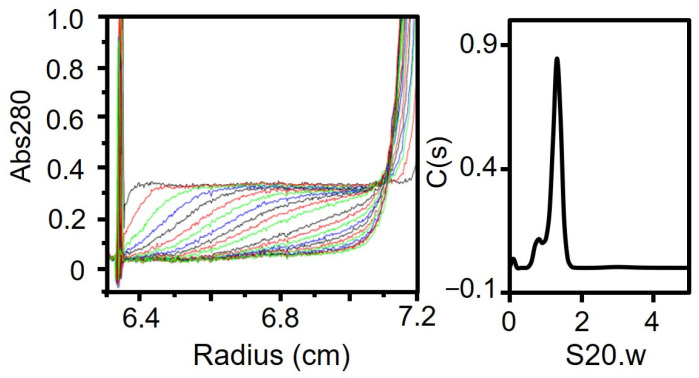
Analytical ultracentrifugation of fragment C. Fragment C was analyzed as described for Figure 2C. The results are summarized in Table 1.

**Figure 6 biomolecules-15-00429-f006:**
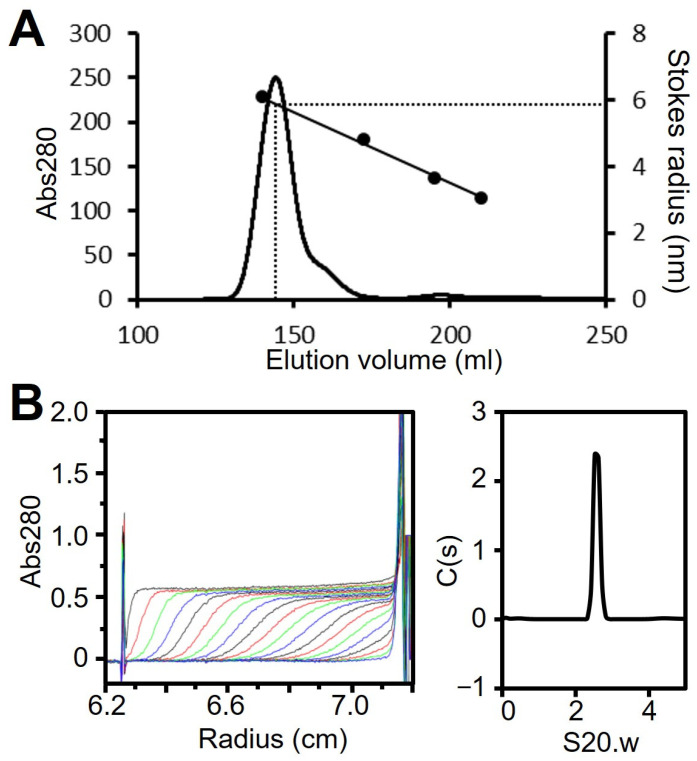
Analysis of fragment D. (**A**) Size exclusion chromatography at a flow rate of 2.5 mL/min. Standard proteins with Stokes radii of 6.1, 4.8, 3.6, and 3.1 nm were used (black dots). (**B**) Analytical ultracentrifugation. Fragment D was analyzed as described for Figure 2C. The results are summarized in Table 1.

**Figure 7 biomolecules-15-00429-f007:**
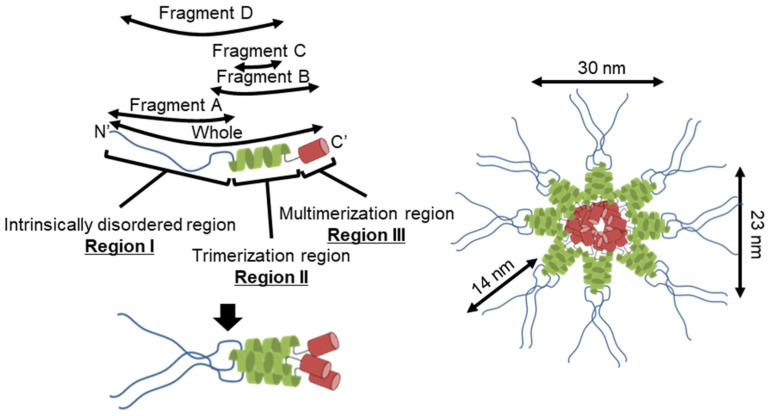
Schematic representation of the structure (**left**) and assembly (**right**) of P65 complexes.

**Figure 8 biomolecules-15-00429-f008:**
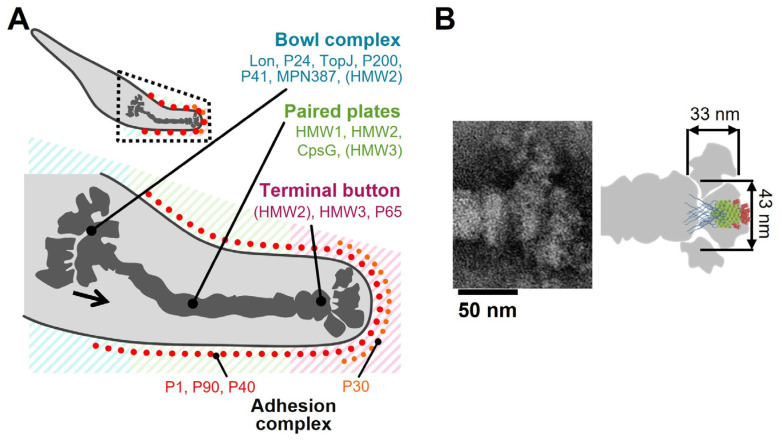
Assignment of P65 onto the image of attachment organelle. (**A**) The whole cell image is shown in the upper panel. The region in dotted lines is magnified in the lower panel, showing the localization of component proteins. The suggested force transmission is indicated by a black arrow. This figure was modified from [21]. (**B**) Schematic representation of P65 on a composite negative staining EM image of the isolated core (modified from Figure 2 of [21]). The P65 is shown as a side view of the bottom image in Figure 7.

**Table 1 biomolecules-15-00429-t001:** Analysis by hydrodynamic parameters.

Fragment	Amino Acid Sequence		Gel Filtration	Analytical Ultracentrifugation				
	Mass (Da)	Stokes Radius as Spherical Monomer (nm)	Stokes Radius (nm)	S20.w	Stokes Radius (nm)	Dimension (nm)	Mass (kDa)	Assembly
Whole	49,203	2.9	n/a	22.2	32.9	n/a	3.10 × 10^3^	Multimer
A	23,868	2.2	3.4	0.97	6.4	48.7 × 1.1	25.7	Monomer
C	8631	1.5	n/a	1.30	5.2	3.8 × 2.4	25.6	Trimer
D	39,080	2.7	5.9	2.56	10.7	21.3 × 2.1	105	Trimer

## Data Availability

The original contributions presented in this study are included in the article/Appendix A. Further inquiries can be directed to the corresponding author.

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
