# Peer review of "Assembly Formation of P65 Protein, Featured by an Intrinsically Disordered Region Involved in Gliding Machinery of Mycoplasma pneumoniae"

_biomolecules, 2025, doi:10.3390/biom15030429_

Round 1
Reviewer 1 Report
Comments and Suggestions for Authors
-
Lack of Functional Validation:
The study focuses on structural analysis without sufficient functional experiments to confirm P65’s role in motility. -
Limited In Vivo Relevance:
In vivo validation is missing, making it unclear how the structural data relates to Mycoplasma pneumoniae’s biological function. -
Clarity and Data Integration Issues:
The connection between experimental data (SAXS, EM) and the proposed P65 model needs clearer presentation for better comprehension.
Author Response
Thank you very much for valuable suggestions.
We agree your comments, and clarified this point in a section “Conclusion” as presented L362- “We clarified the structural outline of P65 protein, a component of internal structure for M. pneumoniae binding and gliding, by using a recombinant protein. In the future, this information would be useful to assign P65 protein into the whole image of the attachment organelle, probably based on cryo EM of a cell and isolated organelle, and discuss the role of P65 protein in binding and gliding mechanisms.”.
Reviewer 2 Report
Comments and Suggestions for Authors
Yabe et al. report biophysical studies on the P65 protein from M. pneumoniae. Reported experiments are thorough and provide a comprehensive view of the different domains of this protein. The description of materials and methods used is appropriate and the results and discussion are straightforward. I have only minor comments to address.
- Insufficient identifiers were provided for the organism and the protein. Please provide the exact strain name and an genbank identifier for the genome entry. For the protein, NCBI number 876932 was given on line 83, which could not immediately be located in the Entrez database. Provide additional identifiers such as an UniProt or Swiss-Prot ID.
- The source of the EM images in Fig. 8b is unclear. It seems that this image was not obtained as part of this study but from previous work. This needs to be clarified and cited (also in the figure legend) if it is from previous work.
Author Response
Thank you very much for your evaluation.
The sequence ID was our mistake. We fixed it and also added strain and original paper as presented L94, “The codons of the p65 gene (AGC04228) from M129-B7 strain [39] were optimized for expression in E. coli BL21(DE3),....”
We also clarified the source of EM images as presented in the legend for Fig. 8, “Schematic representation of P65 on a composite negative staining EM image of the isolated core (modified from Fig. 2 of [20]).”
Reviewer 3 Report
Comments and Suggestions for Authors
In the paper “Assembly formation of P65, an intrinsically disordered protein involved in gliding machinery of Mycoplasma pneumoniae” , the authors carry out a structural and oligomeric state analysis of different fragments of the protein P65.
The paper is interesting, it is well organized; it takes you step by step through the different techniques and conclusions.
I just have some minor concerns.
- The title classifies P65 protein as an intrinsically disordered protein. According to the authors, P65 has ~217 residues that “are likely to be intrinsically disordered” however the rest have “structure”. I don't think this makes P65 an IDP. The title needs to be changed.
- How are the authors sure that the “short peptide, MGSSHHHHHHSSGLVPRGSH”
at the end of the sequence does not affect the shape and conformation results?
Maybe everything would be different if they didn't have this peptide.
- In some places the authors reach conclusive (may be logical) statements, with non-definitive evidence.
For instance: region I is intrinsically disordered because 6 M GdnHCl did not significantly change the CD.
But also, because the CD in the presence of 6 M GdnHCl was different, they conclude that the fragment A has residual secondary structure in solution.
- Would it be possible to test the proposed model of P65 complexes in figure 7. by using SAXS on the whole protein and perform some type of reconstruction shape analysis?
Author Response
Thank you very much for your evaluation.
We modified the title like “Assembly formation of P65 protein, featured by an intrinsically disordered region involved in gliding machinery of Mycoplasma pneumoniae”
Regarding the effects of the tag peptide and the residual structure in disordered region, we discussed as presented, L248- “However, a slight shoulder was observed between 210 and 230 nm. Furthermore, the intensity in this region became closer to zero when 6 M GdnHCl was added. These results suggest that fragment A has residual secondary structure in solution, as has often been observed in intrinsically disordered regions [51]. This structure is unlikely to be derived from the short peptide “MGSSHHHHHHSSGLVPRGSH” fused at the N-terminus, because this sequence is predicted to be fully disordered by ColabFold [52] and has been reported to have disordered structures when fused to a globular protein (PDB ID: 1UPI) [53].”
Regarding the SAXS analysis of the whole complex, we suggest possibilities of cryo EM as presented in Conclusion section L362-, “We clarified the structural outline of P65 protein, a component of internal structure for M. pneumoniae binding and gliding, by using a recombinant protein. In the future, this information would be useful to assign P65 protein into the whole image of the attachment organelle, probably based on cryo EM of a cell and isolated organelle, and discuss the role of P65 protein in binding and gliding mechanisms.”
Round 2
Reviewer 1 Report
Comments and Suggestions for Authors
The article can go for the publication process